# Metagenomic Sequencing of Positive Blood Culture Fluid for Accurate Bacterial and Fungal Species Identification: A Pilot Study

**DOI:** 10.3390/microorganisms11051259

**Published:** 2023-05-10

**Authors:** Edwin Kamau, Shangxin Yang

**Affiliations:** Department of Pathology and Laboratory Medicine, UCLA David Geffen School of Medicine, Los Angeles, CA 90095, USA; edwin.kamau.mil@health.mil

**Keywords:** blood culture, metagenomics, next-generation sequencing, microbial identification, bloodstream infections, mNGS

## Abstract

With blood stream infections (BSIs) representing a major cause of mortality and morbidity worldwide, blood cultures play a crucial role in diagnosis, but their clinical application is dampened by the long turn-around time and the detection of only culturable pathogens. In this study, we developed and validated a shotgun metagenomics next-generation sequencing (mNGS) test directly from positive blood culture fluid, allowing for the identification of fastidious or slow growing microorganisms more rapidly. The test was built based on previously validated next-generation sequencing tests, which rely on several key marker genes for bacterial and fungal identification. The new test utilizes an open-source metagenomics CZ-ID platform for the initial analysis to generate the most likely candidate species, which is then used as a reference genome for downstream, confirmatory analysis. This approach is innovative because it takes advantage of an open-source software’s agnostic taxonomic calling capability while still relying on the more established and previously validated marker gene-based identification scheme, increasing the confidence in the final results. The test showed high accuracy (100%, 30/30) for both bacterial and fungal microorganisms. We further demonstrated its clinical utility especially for anaerobes and mycobacteria that are either fastidious, slow growing, or unusual. Although applicable in only limited settings, the Positive Blood Culture mNGS test provides an incremental improvement in solving the unmet clinical needs for the diagnosis of challenging BSIs.

## 1. Introduction

Bloodstream infections (BSIs) are often associated with severe diseases with high morbidity and mortality [1,2,3,4], making the detection of disease-causing pathogens in BSIs one of the most important functions of the clinical microbiology laboratory [5]. The increased mortality of BSIs is associated with delayed, insufficient, or inappropriate anti-infective treatment [1,3,4]. The accurate and timely identification of BSIs is critical for patient management, infection control, and disease epidemiology [6].

Currently, blood cultures coupled with antimicrobial susceptibility testing (AST) are considered the gold standard for diagnosing BSIs [1,5]. The conventional workflow consists of first subculturing the primary positive blood culture medium to a solid agar plate to obtain pure isolates (colonies) after the blood culture vial is flagged positive, and then the subsequent microbial identification and AST. For many bacteria or yeasts, only 12–48 h of incubation is needed to grow enough microbes for subsequent workups. However, in some instances, bacteria or fungi may grow poorly, slowly, or not grow at all at the subculture step once the primary blood cultures turn positive. In addition, some unusual microbes grown from the subculture of positive blood culture may not be identified by conventional methods such as biochemical tests or mass spectrum-based methods including the matrix-assisted laser desorption/ionization–time of flight (MALDI–TOF). In such situations, the microbial result usually stops short at the descriptive identification based on Gram stain morphology, which is insufficient information for both diagnosis and the empirical treatment of BSIs. This is a clinical dilemma encountered especially in tertiary hospitals, treating many immunocompromised patients who can be infected with uncommon microbes. In such settings, there is an unmet clinical need that warrants a solution. 

In our previous case study, we demonstrated a proof of concept of using a shotgun metagenomics approach to correctly identify an unusual anaerobe, *Catabacter hongkongensis*, directly in a blood culture fluid because the conventional microbial workups including MALDI–TOF failed to identify the bacteria [7].

In the current study, we went further to develop and validate a shotgun metagenomics (mNGS)-based clinical test that can directly identify bacteria and fungi in positive blood culture fluid. This test is intended to be used in situations when conventional laboratory methods for positive blood culture fail to provide an appropriate identification of the microorganism, or a faster turn-around time (TAT) is desired especially for mycobacterial or certain fungal organisms that are typically slow growing. This pilot study is aimed at assessing the accuracy and potential clinical utility of a cutting-edge NGS-based molecular diagnostic assay for BSIs.

## 2. Materials and Methods

### 2.1. Specimen and Conventional Microbiology

Blood or sterile body fluids were collected from patients at the UCLA hospitals as part of routine care and transported to the clinical microbiology laboratory within 4 h, and incubated in the automated BD BACTEC™ FX blood culture system (Becton Dickinson, Franklin Lakes, NJ, USA). These specimens were collected in the following media/bottle types: BACTEC™ Plus Aerobic/F; BACTEC™ Lytic/10 Anaerobic/F; BACTEC™ Peds Plus; BACTEC™ Myco/F Lytic. When vials were flagged positive, they were unloaded and worked up following standard laboratory procedures. Gram stain, fungal stain, modified Ziehl–Neelsen stain or Ziehl–Neelsen staining were performed on the positive blood culture fluid. Specimens with positive stain results were subcultured to the appropriate media and incubated appropriately. For specimens with negative stain results, a blind subculture was also set up, and an acridine orange stain was performed to rule out organisms without cell walls. Isolates were given taxonomic or descriptive identification based on the staining results and one or more of the following methods: biochemical tests (e.g., oxidase, catalase) and/or commercial ID methods including VITEK MS (bioMérieux, Hazelwood, MO, USA), VITEK 2 (bioMérieux), API (bioMérieux), RapID™ ANA II System (Thermo Fisher Scientific, Waltham, MA, USA), PathoDX (Remel, Lenexa, KS, USA), and AccuProbe (Hologic, San Diego, CA, USA). After the standard microbiological workups were performed, 200 µL of each positive blood culture fluid was used for mNGS. A total of 30 samples were included in this study, including 28 randomly selected unique positive blood culture fluid specimens and 2 additional samples from 2 case studies (described in Section 3) due to either failed species identification by the conventional methods, or urgent clinical needs to improve the TAT (Table 1). 

### 2.2. Metagenomic Sequencing

Metagenomic sequencing was performed directly from the blood culture fluid as previously described [6]. Briefly, 200 µL of blood culture material was obtained and DNA was extracted using Qiagen EZ1 Blood and Tissue Kit on the EZ1 Advanced XL instrument per the manufacturer’s recommendations (Qiagen, Valencia, CA, USA). Mycobacterial specimens were first heat-inactivated (100 °C for 30 min) and an additional bead-beating step was performed for the mechanical disruption of the cell wall. Bead-beating step was also performed for fungal specimens. Libraries were prepared and sequenced on the Illumina MiSeq System (Illumina, San Diego, CA, USA) using a 2 × 250 bp standard protocol per the manufacturer’s recommendations. Detailed steps regarding library preparation and sequencing have previously been described in our study on the validation and implantation of whole genome sequence-based bacterial identification in the clinical microbiology laboratory [8]. 

### 2.3. Bioinformatics Analysis 

The data analysis pipeline is illustrated in Figure 1, with 3 major steps including (1) using Chan Zuckerberg ID (CZ-ID; previously known as IDseq) platform (https://czid.org/ (accessed on 1 December 2022)), an open source cloud-based metagenomics pipeline for the initial assessment of the microbial content and selection of candidate species identification; (2) using CLC Genomic Workbench version 12 software (Qiagen, Valencia, CA, USA) to generate consensus sequences for the applicable marker genes including *16S rRNA*, *rpoB, groEL, ITS*, and *28S rRNA* based on the reference genome of the candidate species; and (3) using the NCBI BLAST tool (https://blast.ncbi.nlm.nih.gov/Blast.cgi (accessed on 1 December 2022)) and the Westerdijk Fungal Biodiversity Institute Pairwise Alignment tool (https://wi.knaw.nl/page/Pairwise_alignment (accessed on 1 December 2022)) for bacterial and fungal species level identification, respectively. The sequence (fastq) files generated on MiSeq were uploaded on CZ-ID, which performs host and quality filtration steps, and then executes an assembly-based alignment pipeline which results in the assignment of reads and contigs to taxonomic categories [9]. Based on the outcome of the analysis, the top taxonomies with the highest abundance were chosen as the candidate microorganisms present in the positive blood culture fluid. The subsequent bioinformatics were performed using the validated UCLA NGS bacterial or fungal workflows as previously described [8]. Briefly, for the identification of bacterial species, a reference genome suggested by the CZ-ID results was downloaded directly onto the CLC Genomics Workbench 12 (also referred to as CLCbio), and its three target genes (i.e., *16S rRNA*, *rpoB*, *groEL*[*hsp65*]) were identified, extracted, and concatenated. These genes have been extensively used for bacterial species level identification. The raw sequence reads of the sample were then mapped to the concatenated reference sequence. Various QC metrics were recorded including the total reads, the average coverage, the percentage with at least 5× and 10× coverage, and the percentage of ambiguous nucleotides. The consensus sequence of each target gene was queried using the BLAST NT and/or NR databases. The identification of the bacteria to species or genus level was based on results with at least 99% full-gene coverage, and at least 99% pairwise identity, which calculates the percentage of identical nucleotides in alignment between two sequences. High pairwise identity percentage provides increased confidence that the query sequence is identical to the reference sequences. A cascading taxonomic calling scheme starting from *16S rRNA* first, if not resolved (e.g., two species with the same pairwise identity), then to *rpoB*, and last to *groEL* was used for the species identification (Figure 1). A more detailed algorithm of the scheme was described previously [8]. For the identification of fungal organisms, raw sequences from each sample were processed using the same workflow except that the internal transcribed spacer (ITS) region (encompassing ITS1, 5.8S rDNA, and ITS2) and 28S rRNA gene were used as marker genes [10]. These markers are widely used for the species identification of fungal organisms. Figure 1 provides an overview of the workflow used in the identification of the microorganism in this study.

The organism identification was carried out using the CZ-ID portal pipeline, then confirmed using UCLA NGS bacterial or fungal workflows, which are hereon now referred to as Positive Blood Culture mNGS. The clinical utility of the test is demonstrated by case studies presented here after the implementation of the validated test in our laboratory.

### 2.4. Clinical Utility Assessment and Study Ethics

A retrospective chart review was performed on patients for whom Positive Blood Culture mNGS was performed to assess the clinical utility of the NGS results. This study was reviewed by the UCLA Human Research Protection Program and received an IRB exemption.

## 3. Results

### 3.1. Quality Matrices and Bioinformatics Performance

Overall, a range of 1,215,048–12,781,338 sequence reads was acquired per sample (Table 2 and Table 3). For the CZ-ID portal pipeline, various metrics were obtained with QC criteria established on (1) Phred Quality Score, (2) reads per million (rPM), and (3) percent identity (%id). The Phred Quality Score represents confidence in the assignment of each base call by the sequencer, with a score of 60 representing an accuracy of 99.9999%; all the 28 samples passed the QC cutoff (60%). The rPM assesses the relative abundance of nucleic acid associated with the taxon present in the sample; a high rPM was acquired for all samples (mean= 280,248, range 7534.8–601,308), and thus 5000 rPM was set as the QC cutoff. The %id is the average percent identity of the reads and contigs that were aligned to the taxon in the NCBI NT/NR database; high %id (mean = 98.8, range 83.7–100) was achieved, indicating generally a high confidence in taxonomic calling. Overall, the CZ-ID provided very helpful preliminary microbial species identifications for the downstream analysis.

In the downstream in-house bioinformatics pipelines, high sequence coverage was achieved in 100% (24/24) of the bacterial samples with an average coverage of >20× in all three (*16S*, *rpoB*, and *groEL*) marker genes and <1% ambiguous nucleotide (Table 2). The downstream in-house analysis also identified 6 fungal organisms, with 83.3% (5/6) achieving a high sequence coverage of the ITS region (>20×). However, one *Aspergillus* (UCLA_596) had a lower coverage of 5.9× for ITS, which prompted using an additional marker gene (*28S rRNA*, 9.5× coverage) for species confirmation.

### 3.2. Accuracy

Positive Blood Culture mNGS achieved 100% (30/30) species-level identification, including 24 bacterial and 6 fungal organisms in the positive blood culture fluids (Table 1 and Appendix A). Notably, there was 100% (30/30) concordance between the initial CZ-ID species identifications (the taxonomy with highest abundance) and the final species identifications made by Positive Blood Culture mNGS. In the 24 bacteria, the species identification was all based on the *16S rRNA* gene (>99% pairwise identity) and confirmed by either *rpoB* or *groEL*, except for sample UCLA_478, for which *Slackia exigua* was identified by the *16S rRNA* alone as there were no corresponding *rpoB* and *groEL* reference sequences in the NCBI nt database. All the six fungal organisms were identified based on the ITS region (100% pairwise identity). In sample UCLA_596, in addition to using ITS as the primary marker gene, the species identification (*Aspergillus flavus-oryzae* group) was further confirmed using the *28S rRNA* gene which achieved 100% overlap and similarity.

### 3.3. Case Studies Demonstrating Clinical Utility

Case study 1. A 52-year-old male with a history of rheumatoid arthritis, interstitial lung disease, bilateral orthotopic lung transplant, pulmonary Aspergillosis, *Mycobacterium avium* complex (MAC) infection, abdominal hernia, and pancytopenia had presented with two months of abdominal discomfort and persistent dyspnea that was worsening. The patient reported unintended weight loss and complained of worsening fatigue, brain fog, shortness of breath, poor appetite, and distention of the abdomen. On admission, he was pancytopenic with elevated alkaline phosphatase. Lung imaging studies revealed increased small right pleural effusion with contiguous rounded atelectasis of the right posterior lung base, and ground-glass opacity of the left mid-lung. A liver ultrasound revealed a shrunken nodular liver and moderate to large ascites. A paracentesis was performed with two liters of serous removed. A liver biopsy revealed nodular regenerative hyperplasia and numerous non-caseating granulomas, but without fibrosis or cirrhosis. Bone marrow biopsy also revealed the presence of small non-caseating granulomas. On hospital day 9, blood samples were collected and submitted for blood cultures, and plasma was submitted to Karius^®^ (Redwood City, CA, USA) for a microbial cell-free DNA next-generation sequencing test (KT). The KT results revealed the presence of MAC (254 molecules per microliter). Our laboratory was contacted by the infection prevention team to further speciate the MAC as part of an on-going FDA investigation and continued monitoring of *Mycobacterium intracellulare subsp. chimaera* infections associated with water-based heater-cooler devices (HCD) used during cardiopulmonary bypass in surgeries [11]. The blood culture turned positive after 9 days of incubation, and a blood culture fluid sample (UCLA_1007) was obtained directly from a blood bottle and a Positive Blood Culture mNGS was performed as described. In 96 h, Positive Blood Culture mNGS revealed that the MAC in the patient’s blood was *M*. *avium* subsp. *hominissuis*, thus not related to the HCD-associated *M. intracellulare subsp. chimaera* outbreak, providing a quick subspecies-level identification for the infection prevention and clinical teams to be able to make relevant decisions on the next course of action. The patient was treated and discharged from the hospital and continued to be followed up as an outpatient.

Case study 2. A 76-year-old woman with a history of polymyositis, coronary artery disease, and hypertrophic obstructive cardiomyopathy status post septal myomectomy and mitral valve replacement presented to our emergency department with substernal chest pain and shortness of breath for one day. She also experienced fatigue and generalized weakness for several days and a 25-pound unintentional weight loss over the past year. The preliminary infectious work-up was unrevealing. However, after four days of incubation, aerobic blood cultures collected from her Port-a-Cath grew beaded Gram-positive rods, which were positive for Ziehl–Neelsen acid-fast staining and were reported to be *Mycobacterium* species. Subsequent aerobic blood cultures collected from the Port-a-Cath which were suspected to be the source of the infection grew the same organism. However, MALDI–TOF MS analysis failed to identify these isolates, which prompted us to perform the Positive Blood Culture mNGS from the blood culture bottle (UCLA_505) and we identified the organism as *Mycolicibacterium iranicum*, an emerging rapid-growing mycobacterial pathogen found to infrequently cause opportunistic infections in immunocompromised patients [12,13,14]. With this information, the clinical team optimized treatment and the Port-a-Cath was removed. The patient was successfully treated and discharged.

## 4. Discussion

With BSIs representing a major cause of mortality and morbidity worldwide, blood cultures play a crucial role in diagnosis, but the clinical application is dampened by the long TAT and the detection of only culturable pathogens. In this study, we have developed and validated a shotgun metagenomics test that uses a sample directly from positive blood culture bottles, allowing for the identification of fastidious or slow-growing microorganisms. The test was built based on previously validated WGS tests, which rely on several key marker genes for bacterial and fungal identification. The new test utilizes an open-source metagenomics CZ-ID platform for the initial analysis to generate the most likely candidate species, which is then used as a reference genome for downstream, confirmation analysis. This approach is innovative because it takes advantage of CZ-ID’s agnostic taxonomic calling capability while still relying on the more established and previously validated marker gene-based identification scheme, increasing the confidence in the final results.

In the past decade, pathogen-specific molecular assays targeting the identification of pathogens and some resistance determinants have been developed and tested as alternatives for, or complementary to, blood cultures [15]. Currently, there are several FDA approved tests. Depending on the Gram stain results of positive blood culture, different molecular test panels can be used for the rapid identification of microorganisms [4]. These tests are designed to target a specific set of pathogens and resistance markers, with the number of targets changing and/or increasing as manufacturers improve their technology to meet consumers’ needs. However, these technologies are only as good as the targets included in the tests. As an alternative to the rapid molecular diagnostic panels, metagenomics is now being used for the genomic analysis of specific or all genetic material presented in a clinical sample [4,16]. The sequencing of nucleic acids can be based on specific amplicons (amplicon-based metagenomics) usually restricted to the universal gene targets found in bacteria or fungi [17,18,19] or on the entire genomes (shotgun metagenomics), which allows for the untargeted and unrestricted sequencing of DNA in each sample [16]. Shotgun metagenomics also allows for the detection of resistance markers, virulence genes, and may provide important information on molecular epidemiology [20]. In this study, we have developed and validated a highly accurate shotgun metagenomics-based clinical test for the direct identification of bacteria and fungi from positive blood culture fluids.

Blood cultures are usually incubated for a maximum duration of 5–7 days, with the typical TAT of 48–72 h [21,22,23]. However, prolonged incubation times for fastidious pathogens, mycobacterium, or fungal organisms may be required. Further, blood cultures have reduced sensitivity in patients who have received antibiotics, where there is an inability to culture certain pathogens, and there are requirements for additional testing and wait times for discerning detected pathogens (strain, virulence factors, and antimicrobial resistance) [23,24]. Novel technologies such as syndromic multiplex polymerase chain reaction (PCR) panels, 16S ribosomal DNA (16S rDNA) Sanger sequencing, and MALDI–TOF MS have decreased turnaround times and subsequently have beneficially impacted patient care [25]. Syndromes including pneumonia, meningitis/encephalitis, and sepsis can be caused by a wide array of pathogens with indistinguishable clinical presentations, often necessitating testing with a large panel of diagnostic assays in an attempt to establish the diagnosis. Despite these efforts, the etiology of infectious diseases remains unknown in up to 60% of cases depending on the clinical syndrome [26,27], which results in delayed or ineffective treatments, increased mortality, and excessive healthcare costs. Agnostic shotgun mNGS is emerging as a single, universal pathogen detection method that is revolutionizing infectious disease diagnostics. mNGS allows for the identification and genomic characterization of microorganisms directly from clinical specimens without any a priori information or knowledge of the likely disease causative etiology [28]. Recent workflow improvements and advancements in mNGS technology and bioinformatics tools have reduced barriers such as high costs, long sequencing times, and sophisticated data analysis that have historically made it impractical to apply these technologies routinely in diagnostic laboratories [27,29]. In this study, we have demonstrated clinical utilization of the Positive Blood Culture mNGS test by providing actionable clinical microbiological data in a timely manner from a blood culture specimen that otherwise would not have been possible. We demonstrated the value of this test in an infection control application, enabling the early release of a patient who otherwise would have spent a considerable amount of time in the hospital before *M. chimaera* was ruled out. A recent study demonstrated superior clinical performance (in terms of TAT, sensitivity, and range of organisms identified) of nanopore targeted sequencing compared to blood culture and PCR followed by Sanger sequencing [23]. The clinical utilization of NGS is likely to continue increasing as workflows improve and open-source pipelines such as CZ-ID become more widely available.

Since first being described in 2020, the CZ-ID platform has proven to be uniquely suited for the detection of divergent pathogens [24,30,31,32]. As an open-source cloud-based pipeline, CZ-ID reduces the barrier to entry for mNGS data analysis and enables bench scientists, clinicians, and bioinformaticians to gain insight from mNGS datasets for both known and novel pathogens [9,15]. However, there are two main inherent concerns for using CZ-ID, a cloud-based server for the analysis of sequences that may contain human genomic data. The first concern is uploading human genomic DNA to a cloud server. At the start of this project, we used CLC Genomic Workbench software (Version 12) to interrogate the amount of human genome sequences that may be present in samples sequenced directly from the blood culture bottles, and found that generally human sequence reads were less than 100,000 reads, or <25 million bp (less than 1% of the human genome). The negligible amount of human DNA is due to several reasons. First, a typical blood bottle contains 30 mL of media with 8–10 mL of blood added, which is diluted > 4-fold. Second, the blood bottle is incubated on average 48 h before the growth of bacteria is detected; during this time, blood cells are lysed, and human DNA is rapidly degraded. Third, our method does not sequence very deeply (mean output = 2.9 million reads, maximum output = 12.9 million reads), and thus not many human sequence reads are generated. Last, the CZ-ID pipeline filters out all host genome sequences that are immediately discarded, therefore making sure no human sequences are stored or available for other analyses. Further, all uploaded samples have been completely de-identified. With these factors combined, there is minimal risk that sequences uploaded in the CZ-ID cloud server could compromise a patient’s genetic information. The second inherent concern for using CZ-ID is that this is a research use only tool, and to the best of our knowledge, it has not been validated for diagnosis in the clinical setting. It is important to point out that our assay only uses the CZ-ID as a preliminary assessment for microbial species, mainly to identify a suitable reference genome for the downstream analysis; the definitive identification is based on our laboratory-developed NGS bacterial and fungal bioinformatics workflows. Although these workflows can be used for the identification of microbials without the need for CZ-ID, the use of CZ-ID for the initial analysis simplifies the process and increases confidence in the results obtained, with minimal additional steps in the workflow.

At our institution, we have launched this Positive Blood Culture mNGS test for the diagnosis of BSI. Currently, we perform one NGS run per week with a TAT of 5 to 7 days, which is suboptimal for BSI diagnosis. The main challenges of providing a faster TAT for any NGS test are the cost and labor required to perform it at a higher frequency (e.g., three times per week), which is practically prohibitive in most clinical microbiology laboratories. Significantly reduced cost and full automation of NGS workflow are two essential factors to allow this technology to become a primary identification tool instead of a secondary backup method for only rare scenarios of challenging samples. Furthermore, it should be noted that the Positive Blood Culture mNGS test described in this study has several major differences compared to the plasma cell-free DNA test (e.g., the Karius test). First, the plasma cell-free DNA test does not require blood culture and thus provides faster TAT; however, it also suffers from a poor specificity issue with many false or questionable positive results [33,34]. Conversely, the Positive Blood Culture mNGS test requires incubation time and thus has a slower TAT, but with much higher specificity since the blood culture enriches the microbes to a very high abundance which leads to highly-confident results. Second, due to the limitation of short-read sequencing, the taxonomic calling may not be accurate to the species or the subspecies level when insufficient sequence reads are acquired, which is the common scenario in shotgun metagenomics results. In contrast, this is not a problem for Positive Blood Culture mNGS since large amounts of sequence reads can be obtained to provide sufficient definition for species and even sub-species identification. Due to this, our Positive Blood Culture mNGS is especially useful for mycobacterium species and subspecies identification, which is often critical in the clinical setting as demonstrated in the two case studies described here. In addition, there can be a potential for anti-microbial resistance (AMR) prediction, further justifying utilizing this approach especially for organisms with established AMR genetic markers (e.g., *Mycobacterium tuberculosis*). This is an area we are continuing to explore.

In addition to shotgun mNGS used in this study, pan-bacterial (e.g., based on 16S rRNA) or pan-fungal (e.g., based on ITS) targeted amplicon deep sequencing, also known as targeted NGS (tNGS), is another powerful tool for diagnostic testing [35,36], human microbiome research [37], and ecological microbiome surveillance [38,39]. Compared to shotgun mNGS, tNGS has narrower microbial detection spectrum, but is more cost-effective and less prone to host/background DNA interference, and therefore represents an attractive option for pathogen detection directly from clinical samples. The main limitation of tNGS is its lack of resolution to distinguish certain closely related species; however, clinical validation studies have shown that tNGS identifies most clinically significant pathogens with satisfactory performance [35,36,40].

This study has several limitations. First, the use of a privately funded web portal for the analysis of clinical specimens poses unique challenges: there is no guarantee the portal will always be available, and upgrades or changes may impact its performance. Second, the current TAT for this test is 5–7 days, which is not ideal as discussed above. Third, only a limited species and samples were validated, and no polymicrobial positive blood culture sample was tested in this study. Fourth, similar to blood culture results, caution should be taken to differentiate colonization/contamination from true infection. Importantly, extraneous sources of nucleic acid contamination in the blood bottle medium had been reported to cause false positive results in a commercial direct blood culture molecular identification panel assay (https://www.accessdata.fda.gov/scripts/cdrh/cfdocs/cfRes/res.cfm?ID=181223 (accessed on 21 April 2023); https://asm.org/ASM/media/Policy-and-Advocacy/BlCx-contaminating-DNA-FINAL.pdf (accessed on 21 April 2023)). Therefore, stringent quality control and correlation with other laboratory results (e.g., Gram stain) and clinical presentations are needed to safeguard the accuracy of the test. Due to these limitations, a post-launch test validation is still ongoing.

In conclusion, we developed and validated an innovative NGS test for the accurate species identification directly from the fluid of a positive blood culture bottle, and demonstrated its clinical utility especially for anaerobes and mycobacteria that are either fastidious, slow growing, or unusual. Although applicable only in limited settings, the Positive Blood Culture mNGS test provides an incremental improvement in solving the unmet clinical needs for the diagnosis of challenging BSIs.

## Figures and Tables

**Figure 1 microorganisms-11-01259-f001:**
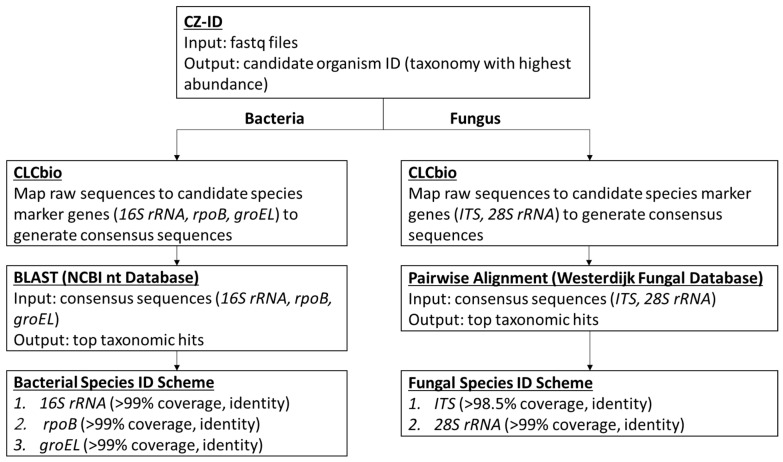
The Bioinformatics Workflow of Positive Blood Culture mNGS.

**Table 1 microorganisms-11-01259-t001:** Microbial organisms identified by Positive Blood Culture mNGS and conventional methods.

Sample ID #	Positive Blood Culture mNGS						Identification by Conventional Methods	Conventional Methods Used for Final Identification
	Identification	Marker Gene Pairwise Identity						
		*16S rRNA*	*rpoB*	*groEL*	*ITS Region*	*28S rRNA*		
* UCLA_467- 473, 562, 558	*Staphylococcus epidermidis*	100.00%	100.00%	100.00%			*S. epidermidis*	MALDI-TOF
UCLA_478	*Slackia exigua*	99.30%	n/a	n/a			*S. exigua*	MALDI-TOF
UCLA_499	*Desulfovibrio diazotrophicus*	99.7%	100.0%	99.9%			*Desulfovibrio* spp.	MALDI-TOF
UCLA_501	*Bacteroides dorei*	99.94%	99.79%	99.94%			*B. dorei*	MALDI-TOF
UCLA_503	*Bacteroides thetaiotaomicron*	99.96%	99.97%	100.00%			*B. thetaiotaomicron*	MALDI-TOF
UCLA_505	*Mycolicibacterium iranicum*	99.87%	97.86%	98.52%			Yellow pigmented rapid-growing *Mycobacterium*	Morphology and Stain
UCLA_510	*Proteus mirabilis*	100.00%	100.00%	100.00%			*P. mirabilis*	MALDI-TOF
UCLA_517	*Escherichia coli*	99.10%	99.98%	100.00%			*E. coli*	MALDI-TOF
UCLA_518	*Staphylococcus lugdunensis*	99.94%	100.00%	100.00%			*S. lugdunensis*	MALDI-TOF
UCLA_519	*Streptococcus intermedius*	99.94%	99.05%	98.34%			*S. intermedius*	MALDI-TOF
UCLA_520	*Clostridium butyricum*	100.00%	99.97%	99.88%			*C. butyricum*	MALDI-TOF
UCLA_524	*Mycobacterium intracellulare subsp. chimaera*	100.00%	100.00%	100.00%			*M. avium* complex	DNA Probe
UCLA_1007	*Mycobacterium avium subsp. hominissuis*	100.00%	100.00%	100.00%			*M. avium* complex	DNA Probe
UCLA_559	*Enterococcus faecium*	99.94%	100.00%	100.00%			*E. faecium*	MALDI-TOF
UCLA_560	*Fusobacterium nucleatum*	100.00%	98.96%	99.88%			*F. nucleatum*	MALDI-TOF
UCLA_561	*Staphylococcus haemolyticus*	99.81%	99.47%	98.40%			*S. haemolyticus*	MALDI-TOF
UCLA_496	*Candida tropicalis*				100%	n/p	*C. tropicalis*	MALDI-TOF
UCLA_497	*Candida glabrata*				100%	n/p	*C. glabrata*	MALDI-TOF
UCLA_498	*Fusarium proliferatum*				100%	n/p	*F. proliferatum*	MALDI-TOF
UCLA_500	*Clavispora lusitaniae*				100%	n/p	*C. lusitaniae*	MALDI-TOF
UCLA_502	*Candida albicans*				100%	n/p	*C. albicans*	MALDI-TOF
UCLA_596	*Aspergillus flavus-oryzae group*				100%	100%	*A. flavus-oryzae group*	MALDI-TOF

* Sample IDs were assigned sequentially to the blood culture bottles in no particular order. There were nine blood bottles that contained *S. epidermidis* with assigned sample IDs UCLA_467 to UCLA_473, UCLA_562, and UCLA_558. n/a: reference sequence not available in the NCBI nt database. n/p: analysis not performed.

**Table 2 microorganisms-11-01259-t002:** Bacterial Identification Quality Control Metrics.

Metric	Range	Average (SD)	QC Criteria
CZ-ID: Phred Quality Score (%) rPM (highest value) %id	65.64–95.33 7534.8–601,308 83.7–100	88.3 (8.5) 280,248 (144,559) 98.8 (3.4)	>60% >5000 >80%
CLC Mapping to Marker Genes: Total Reads Average Coverage (×) 5× Coverage (%)	1,215,048–12,781,338 18.7–352.8 99.2–100	2,853,970 (2,342,278) 109.0 (87.6) 99.9 (0.21)	>1,000,000 >10× >90%
16S rRNA: Length Average Coverage Ambiguous Nucleotide (%)	1513–1573 20.8–3943.7 0.0–0.9	1547.7 (13.5) 942.4 (901.0) 0.08 (0.21)	None >20× <1%
rpoB: Length Average Coverage Ambiguous Nucleotide (%)	3552–4389 30.2–822.4 0.0–0.9	3717.7 (242.0) 176.6 (160.7) 0.05 (0.20)	None >20× <1%
groEL: Length Average Coverage Ambiguous Nucleotide (%)	1620–1647 25.05–717.6 0.0–1.1	1627.0 (9.6) 157.6 (140.8) 0.05 (0.23)	None >20× <1%

**Table 3 microorganisms-11-01259-t003:** Fungal Identification Quality Control Metrics.

Metric	Range	Average (SD)	QC Criteria
CZ-ID: Phred Quality Score (%) rPM (highest value) %id	56.5–90.5 2981.9–619,105.8 96.4–99.8	80.6 (13.6) 225,922.8 (226,785.8) 98.8 (1.2)	>50% >2000 >80%
CLC Mapping to Marker Genes: Total Reads Average Coverage (×) 5× Coverage (%)	2,385,524–4,777,322 5.9–1256.3 66.6–100	3,357,613.7 (779,513.2) 653.2 (556.9) 94.4 (13.7)	>1,000,000 >5× >60%
ITS: Length (nt) Average Coverage (×)	341–2587 5.9–1256.3	880.7 (844.7) 727.7 (569.8)	None >5×
28S (Aspergillus and Mucorales): Length (nt) Average Coverage (×)	593 9.5	593 9.5	None >5×

## Data Availability

The sequencing data of this study is available upon request.

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
