# Peer review of "Metagenomic Sequencing of Positive Blood Culture Fluid for Accurate Bacterial and Fungal Species Identification: A Pilot Study"

_microorganisms, 2023, doi:10.3390/microorganisms11051259_

Round 1

Reviewer 1 Report

The authors developed and validated a shotgun metagenomics test directly on positive blood culture fluid, allowing for identification of fastidious or slow growing microorganisms more rapidly. This work is important, interesting, and provides useful information to researchers in relevant fields. To improve the manuscript, the Discussion should mention that besides the Metagenomic Sequencing, the Amplicon Sequencing (metabarcoding) is also a powerful and widely-adopted method to detect potential bacterial and fungal pathogens, including those in environmental samples (doi: 10.1007/s00253-022-12038-2; 10.1128/Spectrum.00229-21). Although metabarcoding has the advantage of coverage, it is less accurate than metagenomic sequencing.  

Quality of English Language is acceptable.

Author Response

Response: Thank you for your comments. We have included amplicon sequencing based metagenomics (targeted NGS, or tNGS) as another very effective method for direct sample pathogen detection as well as ecological microbiome surveillance (lines 313-320). We also included the 2 suggested papers in the references.

Author Response

The aim of the study was not clearly defined.

Response: We appreciate the reviewer’s comment and opportunity to respond. We have included a clear message stating the aims of this study: “This study is aimed at assessing the accuracy and potential clinical utility of a cutting-edge NGS-based molecular diagnostic assay for BSIs” (lines 63-65).

The Authors wrote that they had developed and validated a shotgun metagenomics test, so what is the purpose of the clinical case reports? It should be explained.

Response: In the Introduction, we explained that the goals of this study include the assessment of the clinical utility. Therefore we used 2 case studies to demonstrate the real-world clinical utility of the test we developed and validated.

The authors wrote that 28 positive blood cultures were tested (line 81), but in the Table 1 there are only 20 species. Have 4 strains been identified as S. epidermidis? If so, there are 24 strains, so 4 are still missing. Nevertheless, this is not clear.

Response: We appreciate the reviewer’s comment. We now see how this can be confusing and we have added the following explanation as a footnote to the table “Sample IDs were assigned sequentially to the blood culture bottles in no particular order. There were nine blood bottles that contained S. epidermidis with assigned sample IDs UCLA_467 to UCLA_473, UCLA_562 and UCLA_558”.

The Authors wrote that they presented the usefulness mainly for anaerobic bacteria and mycobacteria, but only 5 strains of aerobic bacteria and 1 strain of mycobacteria were included in the results.

Response: We appreciate the reviewer’s comments. The blood bottles tested in this study using this methodology were not meant to be exhaustive; this was a test validation study. They represent the organisms that came through our laboratory during the study period that met the inclusion criteria, and we believe this represents what most clinical microbiology laboratories in the US would see where majority of blood stream infections are due to aerobic bacteria. Anaerobic bacteria are less frequently isolated in blood culture, and isolation of mycobacteria from blood culture is even rarer. These are the realistic reasons for the small sample size in this study.

The data in Table 1 are not clear. What was the percentage of identification by conventional methods? Was there only one conventional method for each strain, or more? Were the percentages comparable? What methods were used for each strain? This can be presented in the supplementary material.

Response: We appreciate the reviewer’s comments. Per the title of the table, there are two main columns which contains two methods used for bacterial identification, the positive blood cultures by mNGS and conventional method. In the identification by mNGS, three genes for bacteria and two genes for fungal organisms are used for the identification. As explained in the methods section, bioinformatics analysis, the identification of the bacteria to species or genus level was based on results with at least 99% full-gene coverage and at least 99% pairwise identity (PI). PI is a standard term used in bioinformatics where the alignment of two sequences is compared, the query sequence (user’s input) is compared to the reference sequence to see how the two are closely matched. PI calculates the percentage of identical residues in alignment positions to overlapping alignment positions between the two sequences. The PI between two sequences depends on the alignment and the definition of identity, and alignments vary due to the use of different parameters such as gap penalties and substitution scores. The higher the percentage, the higher the confidence that the reference and query sequences are identical. We have added the following sentence after the 99% pairwise identity (line 117) to make it clearer “which calculates the percentage of identical nucleotides in alignment between the two sequences. High PI percentage provides increased confidence that the query sequence is identical to the reference sequences”.

Reviewer 3 Report

General comments

=============

I appreciated the opportunity to peer-review your work on Metagenomic Sequencing of Positive Blood Culture Fluid for Accurate Bacterial and Fungal Species Identification. This manuscript was well written. 

Specific comments

=============

Major comments

---------------------

1.  In the abstract, on line 16, the abbreviation of the CZ-ID platform should be clarified for the readers. Please provide the full form of CZ-ID and a brief description of its function in the context of this study.

2. The methodology section should include a clearer explanation of the rationale behind selecting 28 samples of positive blood culture. Please elaborate on the duration of blood culture, the random selection process, and how the sample size of 28 was determined. If there was no statistical basis for this sample size, it is recommended to include "pilot study" in the title and manuscript to reflect the preliminary nature of the investigation.

3. In section 3.3, kindly provide a more detailed explanation of the benefits of mNGS into conventional culture studies.

4. In case study 2, the reason for importing a port-a-cath needs to be clarified.

5. There is an inconsistency in Table 1 and case study 2 concerning the identification of Mycolicibacterium iranicum. Please address this discrepancy and explain why it was not included in Table 1, despite being detected in case study 2.

6. Since this study focuses on specimens of positive blood culture, it is essential to address the potential risks of false-positive results or the detection of contaminated pathogens in mNGS. Please add a discussion section outlining these concerns and any precautions taken to minimize them.

7. A comparison of the strengths and weaknesses of the proposed method (mNGS) with other conventional identification methods (such as MALDI-TOF and commercial ID methods) should be provided. Please discuss the advantages and limitations of each method and how they relate to this study's findings.

---------------------

Minor comments

---------------------

8. In the abstract, line 12, please revise the text to maintain consistency in the terminology. Change from "a shotgun metagenomics test (Positive Blood Culture mNGS)" to "a shotgun metagenomics test (mNGS)."

9. On line 106, there is a duplicate URL. Please remove the repeated URL to avoid confusion.

10. In Figure 1, please add explanations for the abbreviations mNGS, CLCbio, and BLAST, to ensure clarity for readers who may be unfamiliar with these terms.

I thought minor editing of English language required.

Author Response

I appreciated the opportunity to peer-review your work on Metagenomic Sequencing of Positive Blood Culture Fluid for Accurate Bacterial and Fungal Species Identification. This manuscript was well written. 

Major comments

---------------------

  1. In the abstract, on line 16, the abbreviation of the CZ-ID platform should be clarified for the readers. Please provide the full form of CZ-ID and a brief description of its function in the context of this study.

Response: We appreciate reviewer’s comment and opportunity to respond. CZ-ID stands for Chan Zuckerberg ID, which was previously named ID-Seq. Unfortunately, there is character limitation in the abstract which doesn’t give us much wiggle room. We understand the reviewer’s concerns and we have added CZ-ID platform link https://czid.org/ in the abstract (line 19). 

  1. The methodology section should include a clearer explanation of the rationale behind selecting 28 samples of positive blood culture. Please elaborate on the duration of blood culture, the random selection process, and how the sample size of 28 was determined. If there was no statistical basis for this sample size, it is recommended to include "pilot study" in the title and manuscript to reflect the preliminary nature of the investigation.

Response: We appreciate the reviewer’s comments. There was no particular rationale for selecting 28 samples, initially this was part of our effort in the attempt to identify organisms when routine laboratory procedures failed to provide adequate or appropriate identification. This doesn’t happen frequently in our laboratory and by the time we reached this number (28), we felt the data collected (over almost 2 years) was sufficient to demonstrate the accuracy and utility of the test. In addition, we mistakenly omitted the 2 samples described in the case studies, and therefore have added them to Table 1. As a result, we now have 30 samples in this study, including 2 more Mycobacteria (updated Table 1).

In addition, we have added “A Pilot Study” in the title, as suggested by the reviewer.

  1. In section 3.3, kindly provide a more detailed explanation of the benefits of mNGS into conventional culture studies.

Response: Thank you for your suggestion, we have added a paragraph (lines 240-252) further explaining the benefits of mNGS.

  1. In case study 2, the reason for importing a port-a-cath needs to be clarified.

Response: We appreciate the reviewer’s comments. In lines 202-203, we have added the following: ‘Port-a-Cath “which was suspected to be the source of the infection” grew the same organisms’. 

  1. There is an inconsistency in Table 1 and case study 2 concerning the identification of Mycolicibacterium iranicum. Please address this discrepancy and explain why it was not included in Table 1, despite being detected in case study 2.

Response: Thank you for pointing out this discrepancy. We mistakenly omitted the 2 samples described in the case studies, and therefore have added them to Table 1. As a result, we now have 30 samples in this study (updated Table 1).

  1. Since this study focuses on specimens of positive blood culture, it is essential to address the potential risks of false-positive results or the detection of contaminated pathogens in mNGS. Please add a discussion section outlining these concerns and any precautions taken to minimize them.

Response: We appreciate the reviewer’s comments. In line 313, we have added these points to the Discussion (lines 325-332).

  1. A comparison of the strengths and weaknesses of the proposed method (mNGS) with other conventional identification methods (such as MALDI-TOF and commercial ID methods) should be provided. Please discuss the advantages and limitations of each method and how they relate to this study's findings.

Response: We appreciate the reviewer’s comment and the opportunity to respond. We have added these points into detailed discussion in lines 243-256.

---------------------

Minor comments

---------------------

  1. In the abstract, line 12, please revise the text to maintain consistency in the terminology. Change from "a shotgun metagenomics test (Positive Blood Culture mNGS)" to "a shotgun metagenomics test (mNGS)."

Response: We appreciate the reviewer’s comment. We have updated this as followings: “shotgun metagenomics next-generation sequencing (mNGS)”

  1. On line 106, there is a duplicate URL. Please remove the repeated URL to avoid confusion.

Response: We appreciate the reviewer’s comment and have fixed this error.

  1. In Figure 1, please add explanations for the abbreviations mNGS, CLCbio, and BLAST, to ensure clarity for readers who may be unfamiliar with these terms.

Response: We appreciate the reviewer’s comments. These terms have been explained in the text including the method section and elsewhere throughout the text.

Reviewer 4 Report

In the present study,the authors  developed and validated a shotgun metagenomics test  directly on positive blood culture fluid, allowing a rsapid detection of fastidious or slow growing microorganisms.There approach is innovative based on an open-source software’s agnostic taxonomic and has clinical utility for fastidious species to be rapidly detected .The authors proceeded to a bioinformatic analysis and yet they showed the clinical utility for fastidious species.

It is a well written paper based on an extensive bibliography .Results are evsluated by bioinformatics in depth followed by a fruitful discussion in comparison with older techniques.

The manuscript will be of high interest to clinicians

My suggestion is to ACCEPT and publish it

Author Response

Response: Thank you very much for your nice comments!

Round 2

Reviewer 2 Report

In the attachment

Author Response

Reviewer 2 (Round 2)

The Authors wrote that they presented the usefulness mainly for anaerobic bacteria and mycobacteria, but only 5 strains of aerobic bacteria and 1 strain of mycobacteria were included in the results.

Response: We would like to clarify that 3 different mycobacteria were included in our study: M. iranicum, M. avium subsp. hominissuis, and M. intracellulare subsp. chimaera.

I suggest removing “mainly” or changing it for “in that”

Response: Searching through our manuscript we didn’t find the word “mainly” that fits the reviewer’s description here. Therefore, we could not make any changes as we don’t really know which exact sentence the reviewer is referring to.

The data in Table 1 are not clear. What was the percentage of identification by conventional methods? Was there only one conventional method for each strain, or more? Were the percentages comparable? What methods were used for each strain? This can be presented in the supplementary material.

In my opinion the questions I asked were not answered. It should be improved. I don’ t know why the Authors don’t want to include the data of conventional identification in the supplementary material. It would be useful for the Readers.

Response: Thank you for this comment. We completely agree that adding more detailed information will help the readers to better understand our data. We have added a column in Table 1 to indicate exactly which conventional method was used to make the final identification. In addition, we also included a Supplemental Table S1 with more microbiology results and morphological descriptions. Please note that for the conventional methods such as MALDI-TOF, there was no percentage to record, because the identifications were called by the commercial system based on its proprietary algorithm and we don’t record any raw results (e.g. scores). Hope this clarifies.